# Neuropeptide VF neurons promote sleep via the serotonergic raphe

Daniel A Lee, Grigorios Oikonomou, Tasha Cammidge, Andrey Andreev, Young Hong, Hannah Hurley, David A Prober*

California Institute of Technology, Division of Biology and Biological Engineering, Pasadena, United States

**Abstract** Although several sleep-regulating neuronal populations have been identified, little is known about how they interact with each other to control sleep/wake states. We previously identified neuropeptide VF (NPVF) and the hypothalamic neurons that produce it as a sleep-promoting system (Lee et al., 2017). Here we show using zebrafish that *npvf*-expressing neurons control sleep via the serotonergic raphe nuclei (RN), a hindbrain structure that is critical for sleep in both diurnal zebrafish and nocturnal mice. Using genetic labeling and calcium imaging, we show that *npvf*-expressing neurons innervate and can activate serotonergic RN neurons. We also demonstrate that chemogenetic or optogenetic stimulation of *npvf*-expressing neurons induces sleep in a manner that requires NPVF and serotonin in the RN. Finally, we provide genetic evidence that NPVF acts upstream of serotonin in the RN to maintain normal sleep levels. These findings reveal a novel hypothalamic-hindbrain neuronal circuit for sleep/wake control.

*For correspondence:
dprober@caltech.edu

Competing interests: The authors declare that no competing interests exist.

## Introduction

While several sleep- and wake-promoting neuronal populations have been identified (*Bringmann, 2018*; *Liu and Dan, 2019*; *Saper and Fuller, 2017*; *Scammell et al., 2017*), fundamental aspects of sleep circuitry organization are poorly understood. Characterizing and understanding the functional and hierarchical relationships between these populations is thus essential for understanding how the brain regulates sleep and wake states (*Oikonomou and Prober, 2017*). Recent evidence from zebrafish and mice demonstrate that the serotonergic raphe nuclei (RN) are critical for the initiation and maintenance of sleep (*Iwasaki et al., 2018*; *Oikonomou et al., 2019*; *Venner et al., 2020*; *Zhang et al., 2018*), in contrast with previous models suggesting a wake-promoting role for the RN that were largely based on their wake-active nature (*Saper and Fuller, 2017*; *Scammell et al., 2017*; *Weber and Dan, 2016*). In zebrafish, mutation of *tryptophan hydroxylase 2* (*tph2*), which is required for serotonin (5-HT) synthesis in the RN, results in reduced sleep, sleep depth, and homeostatic response to sleep deprivation (*Oikonomou et al., 2019*). Pharmacological inhibition of 5-HT synthesis or ablation of the RN also results in reduced sleep. Consistent with a sleep-promoting role for the raphe, optogenetic stimulation of raphe neurons results in increased sleep. Similarly, in mice, ablation of the RN results in increased wakefulness and an impaired homeostatic response to sleep deprivation (*Oikonomou et al., 2019*), whereas chemogenetic stimulation of 5-HT RN neurons (*Venner et al., 2020*) or tonic optogenetic stimulation of 5-HT RN neurons at a rate similar to their baseline pattern of activity (*Oikonomou et al., 2019*) induces sleep. These complementary results in zebrafish and mice (*Oikonomou et al., 2019*; *Venner et al., 2020*), along with classical ablation and pharmacological studies (*Ursin, 2008*), indicate an evolutionarily conserved role for the serotonergic system in promoting vertebrate sleep. However, it is unclear how the RN are themselves regulated to promote sleep.

Viral-tracing studies have identified substantial inputs to the RN from hypothalamic neurons in the lateral hypothalamic area, tuberomammillary nucleus, and dorsomedial nucleus, regions

implicated in sleep-wake regulation (*Pollak Dorocic et al., 2014*; *Ren et al., 2018*; *Weissbourd et al., 2014*). However, it is unknown whether any of these or other populations act upon the RN to promote sleep. One candidate neuronal population expresses the sleep-promoting neuropeptide VF (NPVF) in ~25 neurons in the larval zebrafish hypothalamus (*Lee et al., 2017*). Overexpression of *npvf* or stimulation of *npvf*-expressing neurons results in increased sleep, whereas pharmacological inhibition of NPVF signaling or ablation of *npvf*-expressing neurons results in reduced sleep (*Lee et al., 2017*). While it is unknown how the NPVF system promotes sleep, these neurons densely innervate a region of the hindbrain that is consistent with the location of the RN (*Lee et al., 2017*; *Madelaine et al., 2017*), and NPVF receptors have been shown to be expressed in the RN in zebrafish and rodents (*Bonini et al., 2000*; *Liu et al., 2001*; *Madelaine et al., 2017*; *Roumy et al., 2003*). As perturbations of the NPVF system and RN have similar effects on sleep, *npvf*-expressing neurons appear to project to the RN, and NPVF receptors are expressed in the RN, we hypothesized that the NPVF system promotes sleep via the RN. To test this hypothesis, we explored the relationship between these two neuronal populations using chemogenetics, optogenetics, and calcium imaging. Our results support the hypothesis that the NPVF system promotes sleep via the RN, thus revealing a novel hypothalamus-hindbrain neural circuit for sleep-wake control.

## Results

### NPVF neurons densely innervate the serotonergic inferior raphe

In most vertebrates, the RN are the main source of serotonergic innervation in the brain. In mammals, the RN are divided into two broad nuclei: the superior and inferior raphe nuclei (*Lillesaar et al., 2009*; *Törk, 1990*). The superior nuclei lie on the midbrain/pons boundary (subnuclei B5–B9), and the inferior nuclei in the medulla (subnuclei B1–B3) (*Dahlstroem and Fuxe, 1964*; *Lillesaar et al., 2009*; *Törk, 1990*). Similarly, in zebrafish larvae, developmental studies and neuroanatomical tracings show that the RN are subdivided into the superior raphe (SRa) and inferior raphe (IRa) (*Lillesaar et al., 2009*).

To explore whether the NPVF system may promote sleep via the RN, we first performed a detailed histological analysis of these populations using *Tg(npvf:eGFP)* animals (*Lee et al., 2017*), which specifically label *npvf*-expressing neurons. As previously described (*Lee et al., 2017*; *Madelaine et al., 2017*), the somas of *npvf*-expressing neurons are located in the dorsomedial hypothalamus at 6 days post-fertilization (dpf) (*Figure 1A,B,D*). These neurons send dense and local ramifying projections into the hypothalamus (*Figure 1B,D*), as well as longer range projections into the telencephalon and hindbrain, with a prominent convergence of these projections at the rostral and medial IRa, as confirmed using 5-HT immunohistochemistry (IHC) (*Figure 1B–K* and *Figure 1—figure supplement 1A–B*). These projections form a dense bundle just ventral to the soma of the IRa and also extend dorsally where they appear to make multiple contacts with IRa somas. To confirm this interaction, we mated *Tg(npvf:KalTA4); Tg(UAS:nfsb-mCherry)* (*Agetsuma et al., 2010*; *Lee et al., 2017*) animals, in which NPVF neurons and their processes are labeled with mCherry, to *Tg(tph2:eNTR-mYFP)* animals, in which the SRa and IRa are labeled with membrane-targeted YFP (*Oikonomou et al., 2019*). We observed apparent direct contacts of NPVF neuron fibers with mYFP-labeled IRa soma and fibers (*Figure 1—figure supplement 1C–E*), consistent with a direct interaction between NPVF and IRa neurons.

### Optogenetic stimulation of NPVF neurons results in activation of serotonergic IRa neurons

Based on our histological observations, previous reports that NPVF receptors are present in the RN (*Bonini et al., 2000*; *Liu et al., 2001*; *Madelaine et al., 2017*; *Roumy et al., 2003*), and our demonstration that both NPVF and raphe neurons promote sleep (*Lee et al., 2017*; *Oikonomou et al., 2019*), we hypothesized that NPVF neurons are functionally connected to serotonergic IRa neurons, and that stimulation of NPVF neurons should thus activate IRa neurons. To test this hypothesis, we used *Tg(npvf:ReaChR-mCitrine); Tg(tph2:GCaMP6s-P2A-tdTomato)* animals (*Lee et al., 2017*;

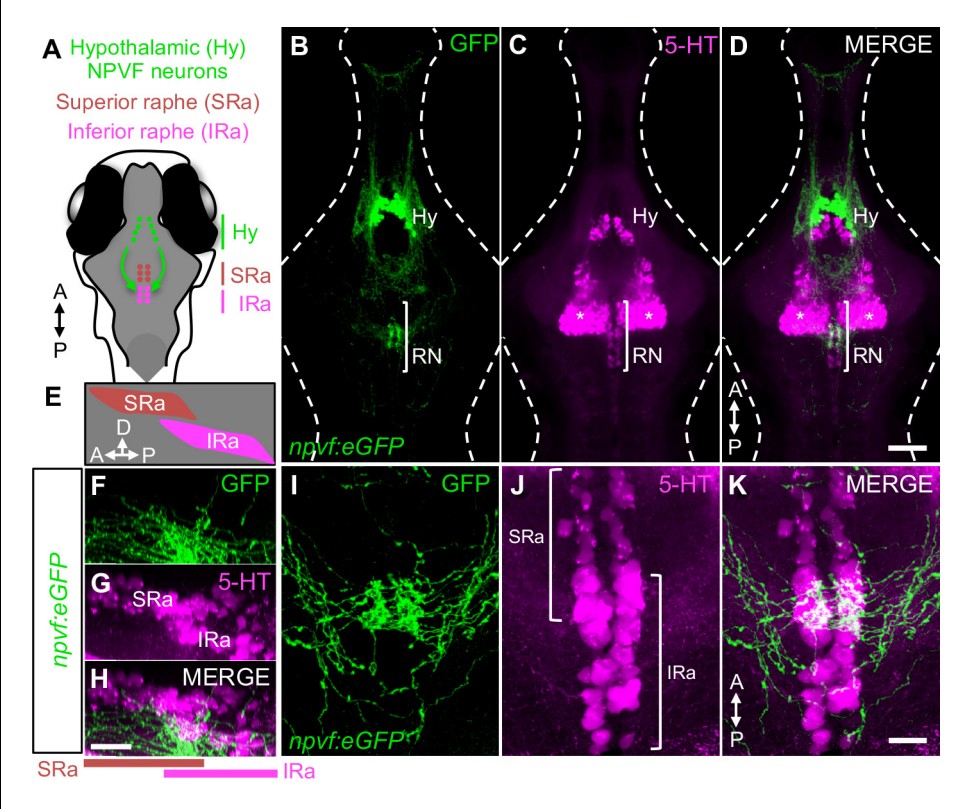

**Figure 1.** Hypothalamic NPVF neurons project to the serotonergic IRa. (A,E) Schematic: 6-dpf zebrafish brain showing location of hypothalamic (Hy) NPVF neurons (green), and the serotonergic superior raphe (SRa, red) and inferior raphe (IRa, magenta). A, anterior; P, posterior; D, dorsal. (B–D) Maximum intensity projection of a brain from a 6-dpf *Tg(npvf:eGFP)* animal (78 μm thick). *npvf*-expressing neurons in the hypothalamus project to the serotonergic raphe nuclei (RN) in the hindbrain (bracket). 5-HT immunohistochemistry labels the RN (bracket), as well as serotonergic populations in the ventral hypothalamus (asterisks) and pretectum. The bracketed region in (B–D) is shown at higher magnification in (I–K) as a maximum intensity projection (50.5 μm thick), with a sagittal view shown in (F–H). Single optical sections are shown in *Figure 1—figure supplement 1*. Scale: 50 μm (B–D), 20 μm (F–H), and 10 μm (I–K).

The online version of this article includes the following figure supplement(s) for figure 1:

**Figure supplement 1.** Projections of NPVF neurons to the serotonergic IRa shown in single optical sections.

*Oikonomou et al., 2019*) to optogenetically stimulate NPVF neurons, while monitoring the activity of IRa neurons. As neurons in the RN are responsive to visible light (*Cheng et al., 2016*), we used invisible 920 nm two-photon light at high-laser power to stimulate ReaChR in NPVF neurons. We also used 920 nm two-photon light, applied at low power, to image GCaMP6s before and after stimulation of NPVF neurons.

To verify that this paradigm indeed results in stimulation of NPVF neurons, we first tested *Tg (npvf:ReaChR-mCitrine); Tg(npvf:GCaMP6s-P2A-tdTomato)* animals (*Lee et al., 2017*; *Lee et al., 2019*). In *Tg(npvf:GCaMP6s-P2A-tdTomato)* animals, *npvf*-expressing neurons express equal levels of GCaMP6s, whose fluorescence intensity serves as a proxy for neuronal activity, and tdTomato (*Lee et al., 2017*). To correct for potential changes in transgene expression or movement artifacts during live imaging, we normalized GCaMP6s fluorescence values to tdTomato fluorescence (for simplicity, hereafter referred to as normalized GCaMP6s fluorescence). We first recorded baseline GCaMP6s and tdTomato fluorescence in *npvf*-expressing neurons, then optogenetically stimulated these neurons, and then recorded post-stimulation fluorescence in these neurons (*Figure 2—figure supplement 1A–L* and *Figure 2—figure supplement 2A,B*). We observed a 92% increase in median normalized GCaMP6s fluorescence in NPVF neurons in *Tg(npvf:ReaChR-mCitrine)* animals (5 animals, 178 neurons) compared to a 4% increase in *Tg(npvf:eGFP)* animals (5 animals, 183 neurons,

p<0.0001, Mann-Whitney test). In *Tg(npvf:ReaChR-mCitrine)* animals, stimulation resulted in increased normalized GCaMP6s fluorescence in nearly all NPVF neurons (*Figure 2—figure supplement 2B*), while there was little or no change in *Tg(npvf:eGFP)* animals (*Figure 2—figure supplement 2A*). Thus, our stimulation paradigm results in robust ReaChR-dependent activation of *npvf*-expressing neurons.

We next used the same stimulation and imaging paradigm to ask whether optogenetic stimulation of NPVF neurons results in activation of serotonergic IRa neurons using *Tg(npvf:ReaChR-mCitrine); Tg(tph2:GCaMP6s-P2A-tdTomato)* animals (*Lee et al., 2017*). To do so, we first recorded normalized GCaMP6s fluorescence in *tph2*-expressing IRa neurons (pre-stimulation), then we stimulated *npvf*-expressing neurons as described above, and then we again recorded normalized GCaMP6s fluorescence in IRa neurons (post-stimulation) (*Figure 2A–H*). Stimulation of NPVF neurons in *Tg(npvf:ReaChR-mCitrine)* animals resulted in a 23% increase in median normalized GCaMP6s fluorescence in IRa neurons (*Figures 2I*; 4 animals, 256 neurons) compared to a 1% decrease in *Tg(npvf:eGFP)* controls (*Figures 2I*; 4 animals, 234 neurons, p<0.0001, Mann-Whitney test). The increased normalized GCaMP6s fluorescence in *Tg(npvf:ReaChR-mCitrine)* animals gradually returned to baseline levels after ~25 s (*Figure 2F,H*), consistent with the prolonged effect expected for neuropeptide/G-protein-coupled receptor (GPCR) signaling (*van den Pol, 2012*). These data suggest that optogenetic stimulation of NPVF neurons results in activation of IRa neurons.

We next analyzed the spatial distribution of IRa neuron responses to stimulation of NPVF neurons. Since the anterior half of the IRa is densely innervated by NPVF neurons but the posterior half is not (*Figure 1* and *Figure 1—figure supplement 1*), we hypothesized that stimulation of NPVF neurons would primarily activate neurons in the anterior IRa. Consistent with this hypothesis, the anterior IRa had significantly more neurons that showed a large increase in normalized GCaMP6s fluorescence (*Figure 2—figure supplement 3C,F,K,M* and *Figure 2—figure supplement 4B*) compared to the posterior IRa (anterior: 50% of 131 neurons; posterior: 18% of 124 neurons, p<0.001, Student's t-test). This pattern of normalized GCaMP6s fluorescence was not observed in the *Tg(npvf:eGFP)* controls (*Figure 2—figure supplement 3A,E,I,M* and *Figure 2—figure supplement 4A*). These results demonstrate that stimulation of NPVF neurons primarily activates neurons in the anterior half of the IRa, consistent with the dense innervation of the anterior IRa by NPVF neurons.

## Loss of *npvf* does not enhance the *tph2* mutant sleep phenotype

The NPVF prepro-peptide is processed to produce three mature neuropeptides, RFRP 1–3 (*Hinuma et al., 2000*). We previously generated zebrafish that contain a frameshift mutation within the *npvf* gene that is predicted to encode a protein that contains RFRP1 but lacks RFRP2 and RFRP3 (*Lee et al., 2017*). We have shown that loss of NPVF signaling due to this mutation (*Lee et al., 2017*), or loss of 5-HT production in the RN due to mutation of *tph2* (*Oikonomou et al., 2019*), results in decreased sleep. Based on our observations that NPVF neurons project to and can activate serotonergic IRa neurons (*Figures 1* and *2*), we next tested the hypothesis that *npvf* and *tph2* act in the same genetic pathway to promote sleep. We tested this hypothesis by comparing sleep in *npvf* -/-; *tph2* -/- animals to their heterozygous mutant sibling controls (*Figure 3*). We reasoned that if *npvf* and *tph2* promote sleep via independent genetic pathways, then animals lacking both genes should sleep more than either single mutant. In contrast, if *npvf* and *tph2* promote sleep in the same pathway, then loss of both genes should not result in an additive sleep phenotype. Similar to our previous results, animals containing a homozygous mutation in either *npvf* or *tph2* slept less than heterozygous mutant sibling controls, with *tph2* mutants showing a stronger phenotype (*Figure 3B, C,E*). However, *npvf* -/-; *tph2* -/- animals did not sleep significantly more than their *npvf* +/-; *tph2* -/- siblings (*Figure 3D,E*), indicating that loss of *npvf* does not enhance the *tph2* mutant phenotype. This result is clear at night but is less clear during the day when *tph2* mutants sleep very little; thus, a potential enhancement of the *tph2* mutant phenotype during the day by loss of *npvf* could be obscured due to a floor effect. Nevertheless, the observation at night is consistent with the hypothesis that *tph2* acts downstream of *npvf* to promote sleep, although we cannot rule out the possibility that *npvf* also promotes sleep through other mechanisms.

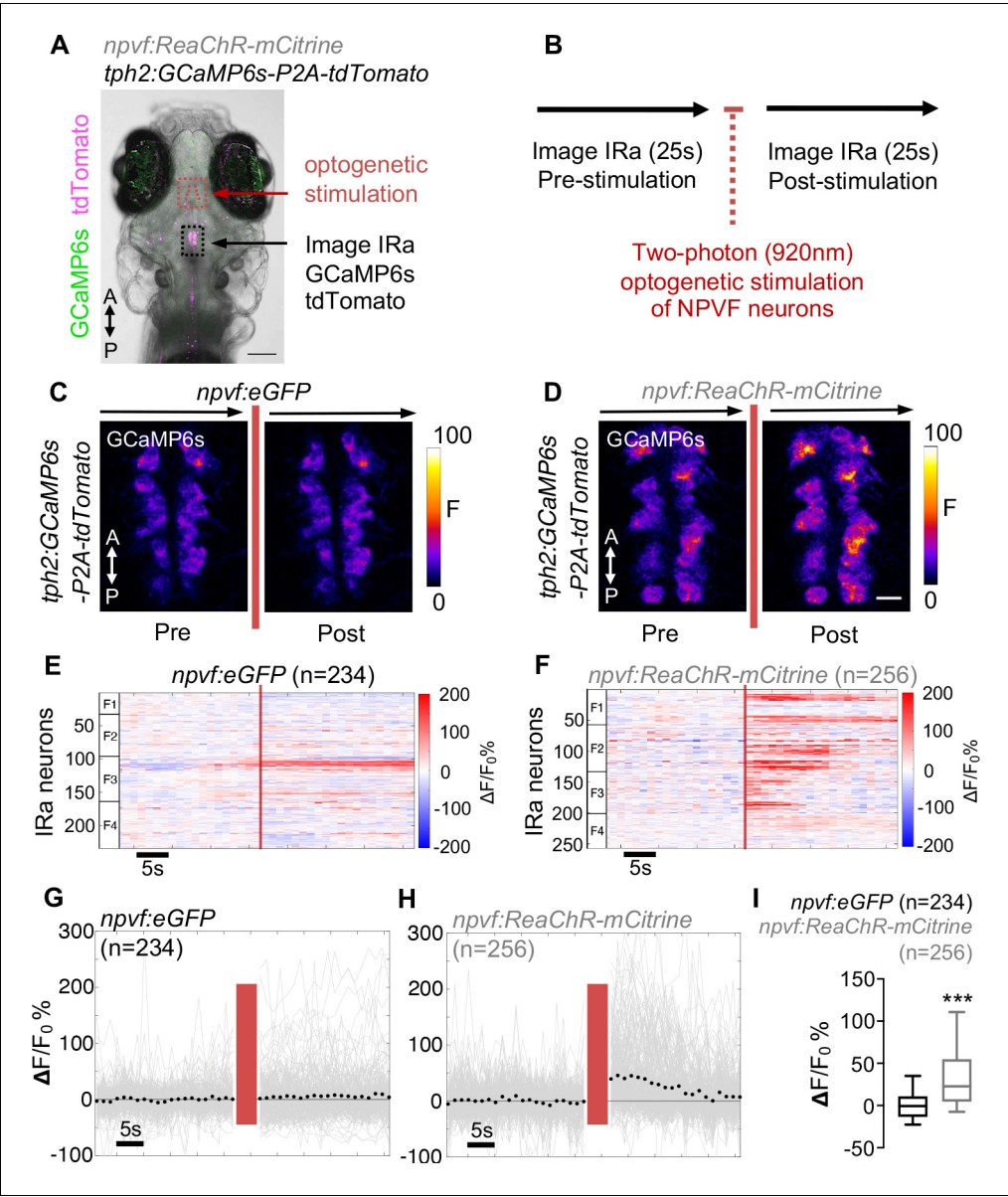

**Figure 2.** Optogenetic stimulation of NPVF neurons activates serotonergic IRa neurons. 6-dpf *Tg(npvf:ReaChR-mCitrine); Tg(tph2:GCaMP6s-P2A-tdTomato)* and *Tg(npvf:eGFP); Tg(tph2:GCaMP6s-P2A-tdTomato)* control animals were analyzed for GCaMP6s/tdTomato fluorescence levels in IRa neurons before and after optogenetic stimulation of NPVF neurons. (**A**) Example of a 6-dpf *Tg(npvf:ReaChR-mCitrine); Tg(tph2:GCaMP6s-P2A-tdTomato)* animal showing GCaMP6s and tdTomato fluorescence in IRa neurons with overlayed NPVF neurons (red circles, different z-plane) indicating region of optogenetic stimulation (red box). Black box indicates region where IRa neurons were imaged and analyzed in subsequent panels. Green and magenta signals in the eyes are due to autofluorescence. (**B**) Normalized GCaMP6s fluorescence in the IRa was first recorded for 25 s (20 frames). NPVF neurons were then optogentically stimulated for 3.7 s (red line), and normalized GCaMP6s fluorescence in the IRa was then immediately imaged again for 25 s. (**C,D**) Average GCaMP6s fluorescence in IRa neurons for 10 imaging frames before (Pre) and after (Post) optogenetic stimulation of NPVF neurons in *Tg(npvf:eGFP); Tg(tph2:GCaMP6s-P2A-tdTomato)* (**C**) and *Tg(npvf:ReaChR-mCitrine); Tg(tph2:GCaMP6s-P2A-tdTomato)* (**D**) animals. (**E–H**) Normalized GCaMP6s fluorescence of individual IRa neurons shown as heat maps (**E,F**), in which each horizontal line represents an IRa neuron, and as line graphs (**G,H**) showing individual (gray lines) and mean (dotted line) IRa neuron responses before and after optogenetic stimulation (red lines) in *Tg(npvf:eGFP); Tg(tph2:GCaMP6s-P2A-tdTomato)* control (**E,G**) and *Tg(npvf:ReaChR-mCitrine); Tg(tph2:GCaMP6s-P2A-tdTomato)* (**F,H**) animals. F1-F4 in (**E,F**) indicate neurons from four different fish. (**I**) Box plot of normalized GCaMP6s $\Delta F/F_0$ values from the average of the first 10 imaging frames post-stimulation of each neuron for *Tg(tph2:GCaMP6s-P2A-tdTomato)* animals that

*Figure 2 continued on next page*

*Figure 2 continued*

also contain either a *Tg(npvf:eGFP)* (black) or *Tg(npvf:ReaChR-mCitrine)* (gray) transgene. n = number of neurons quantified from four animals of each genotype. ***p<0.001, Mann-Whitney test. Scale: 100 μm (**A**), 10 μm (**D**). The online version of this article includes the following figure supplement(s) for figure 2:

**Figure supplement 1.** Validation of two-photon-induced optogenetic stimulation of NPVF neurons.
**Figure supplement 2.** Responses of individual NPVF neurons to optogenetic stimulation of NPVF neurons in individual fish.
**Figure supplement 3.** Optogenetic stimulation of NPVF neurons activates anterior IRa neurons.
**Figure supplement 4.** Responses of individual IRa neurons to optogenetic stimulation of NPVF neurons in individual fish.

## Sleep induced by chemogenetic or optogenetic stimulation of *npvf*-expressing neurons requires serotonergic RN neurons

We next tested the hypothesis that NPVF neuron-induced sleep requires serotonergic RN neurons (*Figure 4A*). To do so, we utilized two approaches to stimulate *npvf*-expressing neurons in freely behaving animals in which the RN were chemogenetically ablated using enhanced nitroreductase (eNTR) (*Mathias et al., 2014*; *Tabor et al., 2014*). eNTR converts the inert pro-drug metronidazole (MTZ) into a cytotoxic compound that causes cell-autonomous death (*Curado et al., 2007*; *Mathias et al., 2014*; *Tabor et al., 2014*). We previously showed that ablation of RN neurons in *Tg (tph2:eNTR-mYFP)* animals results in significantly decreased sleep (*Oikonomou et al., 2019*). This phenotype is similar to those of both *tph2 -/-* zebrafish and to mice in which the dorsal and median serotonergic RN are ablated (*Oikonomou et al., 2019*). Similar to our previous report (*Oikonomou et al., 2019*), treatment of *Tg(tph2:eNTR-mYFP)* animals with 5 mM MTZ during 2–4 dpf resulted in near complete loss of YFP fluorescence and 5-HT immunoreactivity in the RN (*Figure 4B,E,F*). In contrast, treatment of these animals with DMSO vehicle control (*Figure 4C,D*), or treatment of *Tg(tph2:eNTR-mYFP)* negative siblings with MTZ (*Figure 4G,H*), did not cause loss of the RN.

Having confirmed our ability to chemogenetically ablate the RN, we next combined this perturbation with chemogenetic stimulation of NPVF neurons and asked whether the sedating effect of stimulating NPVF neurons is diminished in RN-ablated animals (*Figure 4A,E,F*). Specifically, we expressed the rat capsaicin receptor TRPV1 (*Chen et al., 2016*) in NPVF neurons using *Tg(npvf: KalTA4); Tg(UAS:TRPV1-tagRFP-T)* animals (*Lee et al., 2017*). We previously showed that treating these animals with 2 μM capsaicin (Csn), a TRPV1 small molecule agonist, results in *c-fos* expression in NPVF neurons and increased sleep at night (*Lee et al., 2017*). Following MTZ treatment of animals that do or do not carry the *Tg(tph2:eNTR-mYFP)* transgene, we treated *Tg(npvf:KalTA4)* and *Tg(npvf:KalTA4); Tg(UAS:TRPV1-tagRFP-T)* siblings with Csn. Consistent with our previous observation (*Lee et al., 2017*), Csn-treated *Tg(npvf:KalTA4); Tg(UAS:TRPV1-tagRFP-T)* animals showed a 14% increase in nighttime sleep compared to their Csn treated *Tg(npvf:KalTA4)* siblings in *Tg(tph2: eNTR-mYFP)* negative animals (*Figure 4I–N*), indicating that chemogenetic stimulation of NPVF neurons results in increased nighttime sleep in animals with an intact RN. In contrast, there was no significant difference in nighttime sleep between Csn-treated *Tg(npvf:KalTA4); Tg(UAS:TRPV1-tagRFP-T)*; *Tg(tph2:eNTR-mYFP)* animals and their Csn-treated *Tg(npvf:KalTA4); Tg(tph2:eNTR-mYFP)* siblings (*Figure 4N*). This result indicates that sleep induced by chemogenetic stimulation of NPVF neurons requires the RN.

Similar to chemogenetic stimulation of NPVF neurons, we previously showed that optogenetic stimulation of NPVF neurons results in activation of NPVF neurons and increased sleep (*Lee et al., 2017*). Thus, as an alternative approach to test the hypothesis that NPVF neurons promote sleep via the serotonergic RN, we used *Tg(npvf:ReaChR-mCitrine); Tg(tph2:eNTR-mYFP)* animals to test whether the sedating effect of optogenetic stimulation of NPVF neurons is diminished in RN-ablated animals. To do so, we used a previously described non-invasive, large-scale assay that allows optogenetic stimulation of genetically specified neurons while monitoring up to 96 freely behaving animals (*Singh et al., 2015*). We first recorded baseline behavior for 30 min in the dark, and then exposed the animals to blue light for 30 min. Similar to chemogenetic stimulation, optogenetic stimulation of NPVF neurons in *Tg(npvf:ReaChR-mCitrine)* animals resulted in a 25% decrease in

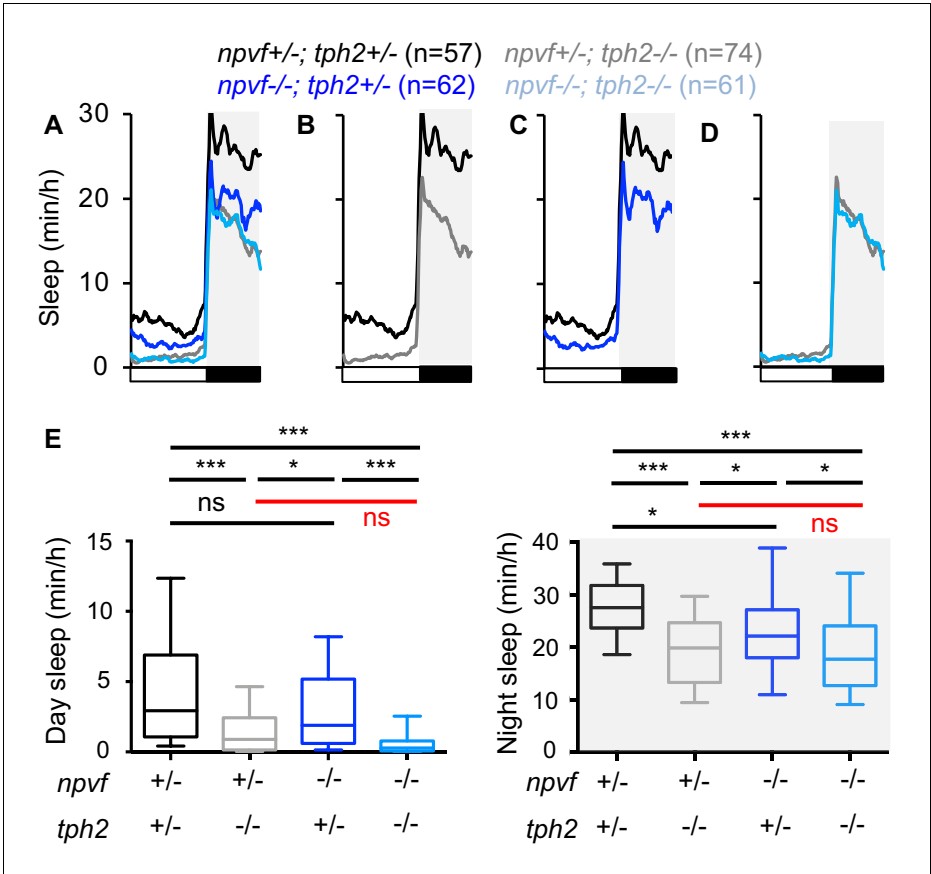

**Figure 3.** Loss of *npvf* does not enhance the *tph2* mutant sleep phenotype. (**A–D**) Sleep for *npvf +/-*; *tph2 +/-* (black), *npvf -/-*; *tph2 +/-* (dark blue), *npvf +/-*; *tph2 -/-* (gray), and *npvf -/-*; *tph2 -/-* (light blue) siblings. The graph on the left shows data for all four genotypes. The three other graphs show the same data as separate pair-wise comparisons. (**E**) Box plots quantify sleep during the day (left) and night (right). White boxes indicate day. Black boxes and gray shading indicate night. Data during the fifth day and night of development from four experiments combined is shown. n = number of animals. ns p>0.05, *p<0.05, ***p<0.005, Two-way ANOVA with Holm-Sidak test for each indicated pair-wise comparison. The comparison in red indicates no significant difference between *npvf +/-*; *tph2 -/-* and *npvf -/-*; *tph2 -/-* siblings.

locomotor activity and a 58% increase in sleep compared to non-transgenic sibling controls (*Figure 5A*). In contrast, following ablation of the RN by MTZ treatment, blue light exposure did not result in a significant difference between the behavior of *Tg(npvf:ReaChR-mCitrine)*; *Tg(tph2:eNTR-mYFP)* animals and their *Tg(tph2:eNTR-mYFP)* sibling controls (*Figure 5B*). Thus, similar to chemogenetic stimulation of NPVF neurons, sleep induced by optogenetic stimulation of NPVF neurons requires the RN. Together, these results are consistent with the model that NPVF neurons act upstream of RN neurons to promote sleep.

## Sleep induced by stimulation of *npvf*-expressing neurons requires *npvf*

The above results and our previous observations (*Lee et al., 2017*) demonstrate that stimulation of NPVF neurons results in increased sleep in zebrafish. However, it is unknown whether this phenotype is due to the action of NPVF or to other factors within these cells, such as the fast neurotransmitter glutamate (*Lee et al., 2017*). To directly test the hypothesis that stimulation of *npvf*-expressing neurons promotes sleep due to release of NPVF, we optogenetically stimulated these neurons in *npvf* mutant animals. Similar to previous results using animals that are homozygous wild-type for *npvf* (*Lee et al., 2017*), optogenetic stimulation of NPVF neurons in *Tg(npvf:ReaChR-mCitrine)*; *npvf +/-* animals resulted in a 25% decrease in locomotor activity and a 28% increase in sleep compared to non-transgenic *npvf +/-* sibling controls (*Figure 5—figure supplement 1A*). In contrast, there was

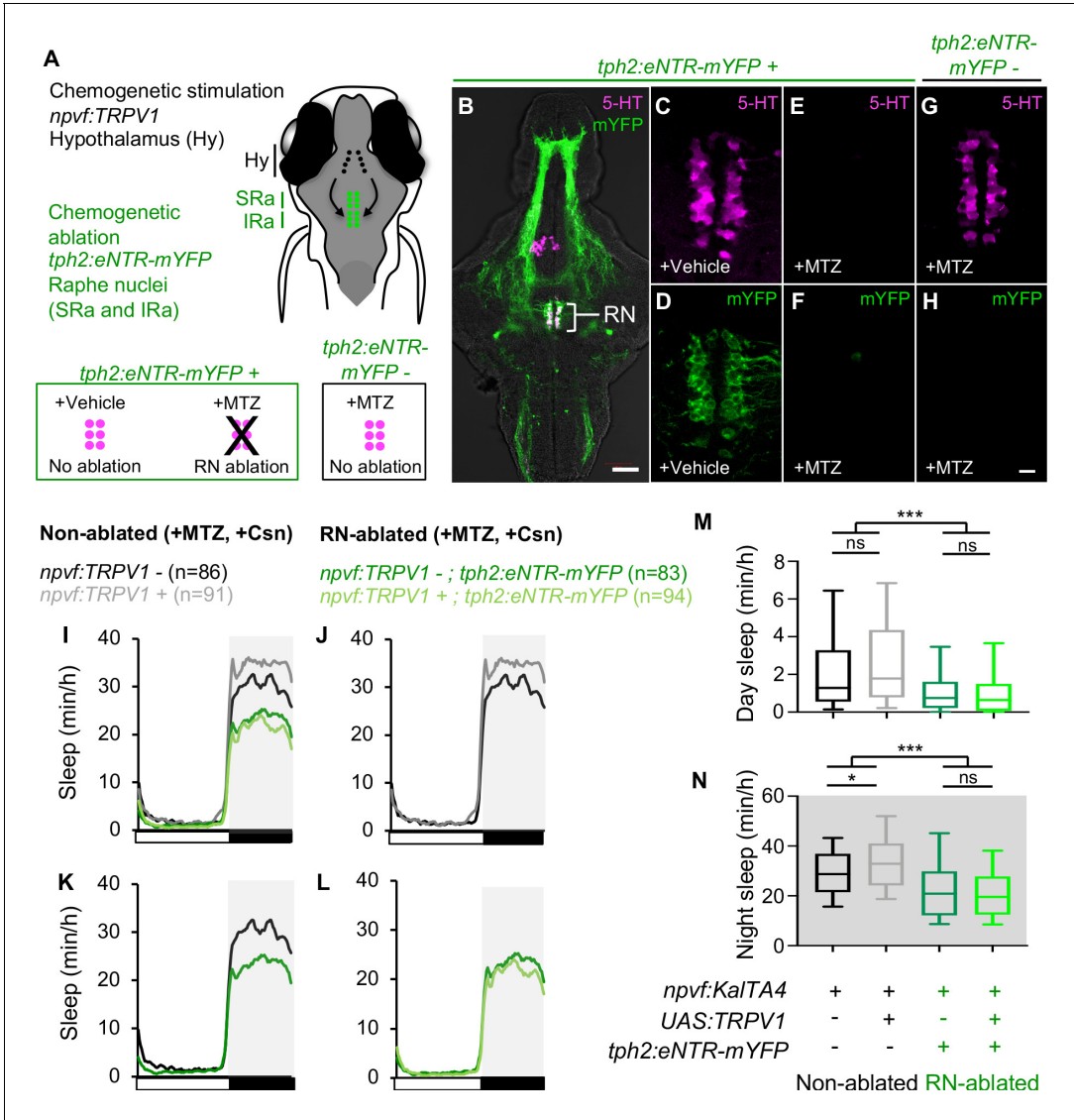

**Figure 4.** Chemogenetic ablation of the RN abolishes sleep induced by chemogenetic stimulation of NPVF neurons. (A) Schematic of experiment. Animals were first treated with MTZ to ablate serotonergic RN neurons in *Tg(tph2:eNTR-mYFP)* animals but not in non-transgenic sibling controls. Behavior was then monitored during chemogenetic stimulation of *npvf*-expressing neurons in *Tg(npvf:KalTA4); Tg(UAS:TRPV1-TagRFP-T)* animals in parallel with their non-stimulated *Tg(npvf:KalTA4)* sibling controls. (B) 5-dpf *Tg(tph2:eNTR-mYFP)* zebrafish brain showing serotonergic RN neurons and some of their projections (green) and labeled with a 5-HT-specific antibody (magenta). The bracketed region is magnified in (C–H). Treatment of *Tg (tph2:eNTR-mYFP)* animals with MTZ results in the loss of both 5-HT immunoreactivity (E) and mYFP (F) in the RN, but treatment with DMSO vehicle control does not (C,D). MTZ treatment of *Tg(tph2:eNTR-mYFP)* negative siblings does not result in loss of RN neurons (G). Images are single 4-µm- (B) and 0.6-µm- (C–H) thick optical sections. Scale: 50 µm (B), 10 µm (C–H). (I–L) Sleep of 5-dpf *Tg(npvf:KalTA4)* (black), *Tg(npvf:KalTA4); Tg(UAS:TRPV1-TagRFP-T)* (gray), *Tg(npvf:KalTA4); Tg(tph2:eNTR-mYFP)* (dark green), and *Tg(npvf:KalTA4); Tg(UAS:TRPV1-TagRFP-T); Tg(tph2:eNTR-mYFP)* (light green) siblings treated with 2 µM Csn. White and black bars under behavioral traces indicate day and night, respectively. (M,N) Box plots quantify sleep during day (M) and night (N). n = number of animals. ns p>0.05, *p<0.05, ***p<0.005, Two-way ANOVA with Holm-Sidak test.

no significant difference between the behavior of *Tg(npvf:ReaChR-mCitrine); npvf -/-* animals and their non-transgenic *npvf -/-* siblings (*Figure 5—figure supplement 1B*). This result indicates that sleep induced by stimulation of NPVF neurons requires NPVF, suggesting that the phenotype is due to NPVF neuropeptide/GPCR signaling. This possibility is consistent with the slow decay of the increased activity of RN neurons following stimulation of NPVF neurons (*Figure 2F,H*, *Figure 2—figure supplement 3F,K* and *Figure 2—figure supplement 4B*).

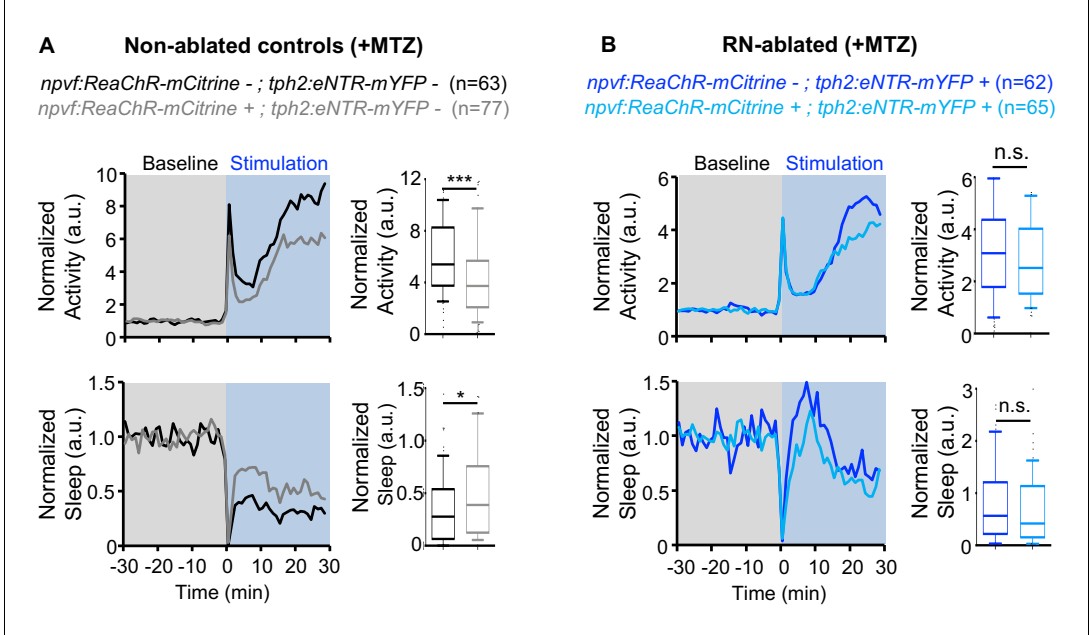

**Figure 5.** Chemogenetic ablation of the RN abolishes sleep induced by optogenetic stimulation of NPVF neurons. Normalized locomotor activity (top) and sleep (bottom) of 5-dpf *Tg(npvf:ReaChR-mCitrine)* (gray and light blue) and non-transgenic sibling control (black and blue) animals before (Baseline) and during blue light exposure (Stimulation) in *Tg(tph2:eNTR-mYFP)* negative (**A**) or positive (**B**) siblings. Because the animals see the blue light, they exhibit a brief startle at light onset that is excluded from analysis, followed by a gradual increase in activity that plateaus after ~15 min. Box plots quantify locomotor activity and sleep for each animal during optogenetic stimulation normalized to the baseline of all animals of the same genotype. n = number of animals. ns p>0.05, *p<0.05, ***p<0.005, Mann-Whitney test.

The online version of this article includes the following figure supplement(s) for figure 5:

**Figure supplement 1.** Sleep induced by optogenetic stimulation of NPVF neurons is abolished in *npvf* mutant animals.

## Sleep induced by stimulation of *npvf*-expressing neurons requires 5-HT in RN neurons

Zebrafish RN neurons produce not only 5-HT, but also other factors such as GABA (*Kawashima et al., 2016*) that may mediate sleep induced by stimulation of NPVF neurons. To distinguish between these possibilities, we tested the hypothesis that NPVF neuron-induced sleep requires the presence of 5-HT in RN neurons. To do so, we used a chemogenetic approach to stimulate NPVF neurons while testing if 5-HT in RN neurons is required for NPVF-induced sleep. Specifically, we compared the effects of chemogenetic stimulation of *npvf*-expressing neurons using TRPV1/Csn in *tph2* -/- animals to *tph2* +/- sibling controls (*Figure 6A–F*). Treatment of *Tg(npvf: KalTA4); Tg(UAS:TRPV1-tagRFP-T); tph2* +/- animals with 2 µM Csn resulted in a 12% increase in nighttime sleep compared to their identically treated *Tg(npvf:KalTA4); tph2* +/- control siblings (*Figure 6A,C,F*). However, there was no significant difference in nighttime sleep between Csn-treated *Tg(npvf:KalTA4); Tg(UAS:TRPV1-tagRFP-T); tph2* -/- animals and their identically treated *Tg (npvf:KalTA4); tph2* -/- siblings (*Figure 6D,F*), suggesting that 5-HT in RN neurons is required for sleep that is induced by NPVF neurons.

As an alternative approach to test the hypothesis that NPVF neuron-induced sleep requires 5-HT in raphe neurons, we compared the effect of optogenetic stimulation of NPVF neurons in *tph2* -/- animals to *tph2* +/- sibling controls (*Figure 6G,H*). Optogenetic stimulation of NPVF neurons in *Tg (npvf:ReaChR-mCitrine); tph2* +/- animals resulted in a 27% decrease in locomotor activity and a 60% increase in sleep (*Figure 6G*). In contrast, there was no significant difference between the behavior of *Tg(npvf:ReaChR-mCitrine); tph2* -/- animals and their non-transgenic *tph2* -/- siblings (*Figure 6H*). Thus, both the chemogenetic and optogenetic stimulation results are consistent with the hypothesis that NPVF neuron-induced sleep requires 5-HT in RN neurons.

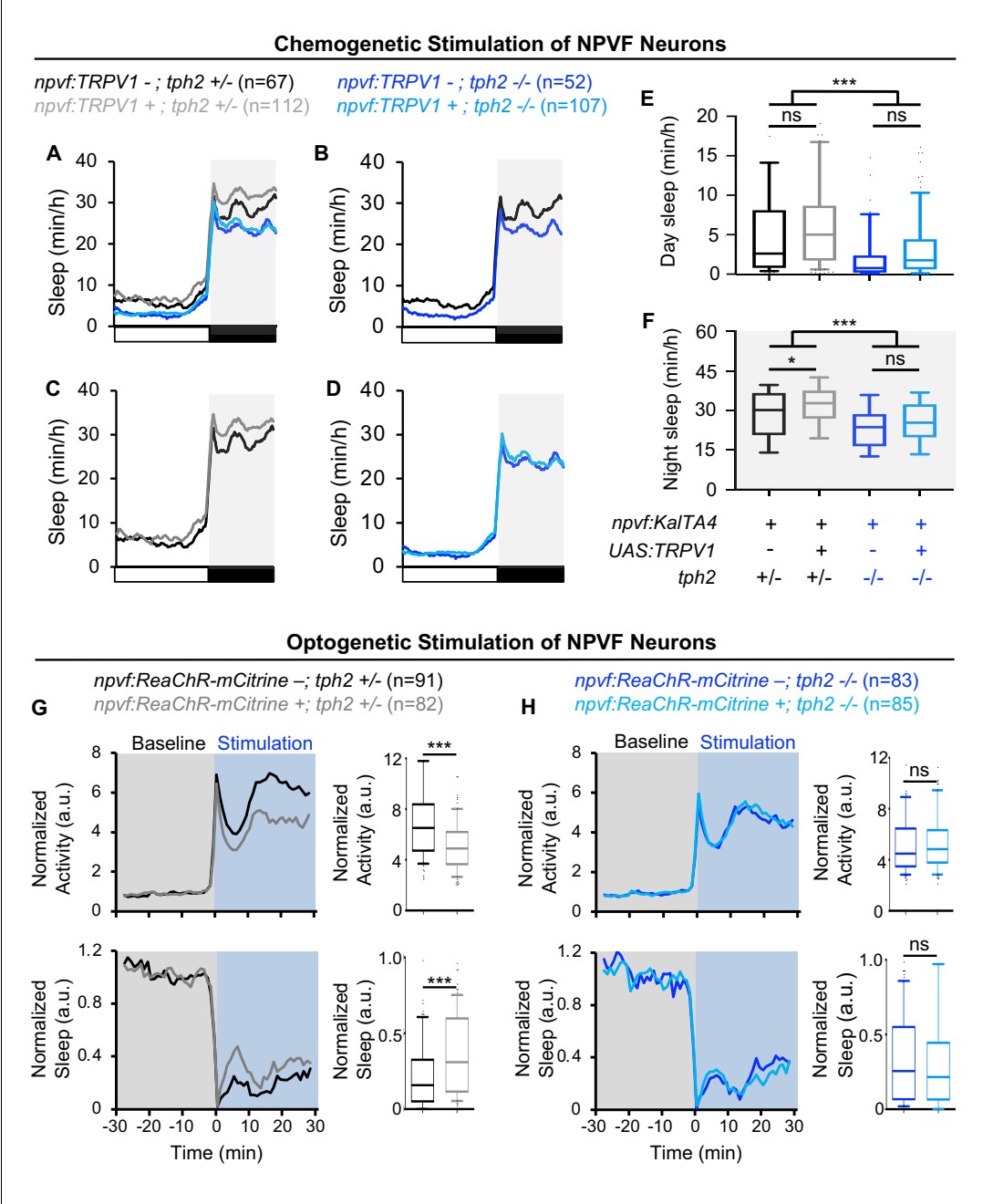

**Figure 6.** Sleep induced by chemogenetic or optogenetic stimulation of NPVF neurons is abolished in *tph2* mutant animals. (**A**) Sleep of 5-dpf *Tg(npvf: KalTA4); tph2+/-* (black), *Tg(npvf:KalTA4); tph2-/-* (dark blue), *Tg(npvf:KalTA4); Tg(UAS:TRPV1-TagRFP-T); tph2+/-* (gray), and *Tg(npvf:KalTA4); Tg(UAS: TRPV1-TagRFP-T); tph2-/-* (light blue) siblings treated with 2 µM Csn. White and black bars under behavioral traces indicate day and night, respectively. (**E,F**) Box plots quantify sleep during the day (**E**) and night (**F**). Chemogenetic stimulation of NPVF neurons increases sleep at night compared to non-transgenic sibling controls in *tph2+/-* animals (**C,F**) but not in *tph2-/-* siblings (**D,F**). n = number of animals. ns p>0.05, *p<0.05, ***p<0.005, Two-way ANOVA with Holm-Sidak test. (**G,H**) Normalized locomotor activity (top) and sleep (bottom) of *Tg(npvf:ReaChR-mCitrine)* (gray and light blue) and non-transgenic sibling control (black and dark blue) animals before (Baseline) and during exposure to blue light (Stimulation) in (**G**) *tph2+/-* or (**H**) *tph2-/-* animals. Box plots quantify locomotor activity and sleep for each animal during optogenetic stimulation normalized to the baseline of all animals with the same genotype. Optogenetic stimulation of NPVF neurons decreases locomotor activity and increases sleep compared to non-transgenic sibling controls in *tph2+/-* animals (**G**) but not in *tph2-/-* siblings (**H**). n = number of animals. ns p>0.05, ***p<0.005, Mann-Whitney test.

## Discussion

The serotonergic RN were first implicated in sleep-wake regulation over 50 years ago, but it has long been disputed whether they act to promote sleep or wakefulness (*Ursin, 2008*). We and others recently addressed this controversy in both mice and zebrafish by providing both gain- and loss-of-function evidence that the serotonergic RN promote sleep (*Iwasaki et al., 2018*; *Oikonomou et al., 2019*; *Venner et al., 2020*; *Zhang et al., 2018*). Our findings agree with invertebrate studies which showed that 5-HT signaling promotes sleep in *Drosophila* (*Qian et al., 2017*; *Yuan et al., 2006*). However, while 5-HT plays an evolutionarily conserved role in promoting sleep, the neuronal mechanism that acts upon serotonergic neurons to promote sleep was unknown. Here we show that *npvf*-expressing neurons in the dorsomedial hypothalamus, which are sleep-promoting (*Lee et al., 2017*), densely innervate and can activate anterior serotonergic IRa neurons, and also require 5-HT in RN neurons in order to induce sleep. Furthermore, our optogenetic and functional imaging data suggest that sleep induced by NPVF neurons is due to NPVF neuropeptide/GPCR signaling. Taken together with observations of NPVF receptor expression in the zebrafish and rodent RN (*Bonini et al., 2000*; *Liu et al., 2001*; *Madelaine et al., 2017*; *Roumy et al., 2003*), these results describe a hypothalamus-hindbrain sleep-promoting neuronal circuit arising from the dorsomedial hypothalamus, a region previously linked to circadian regulation of wakefulness (*Chou et al., 2003*; *Gooley et al., 2006*; *Mieda et al., 2006*), but not to sleep. While we cannot rule out the possibility that NPVF neurons may promote sleep in part through additional mechanisms, our optogenetic, chemogenetic, and genetic epistasis data indicate that most, if not all, of the sleep-promoting properties of NPVF neurons are mediated by the serotonergic RN.

Consistent with the hypothesis that NPVF neurons promote sleep via the RN, we found that stimulation of NPVF neurons results in activation of most serotonergic IRa neurons, especially those in the anterior IRa, which is densely innervated by NPVF neurons. This observation contrasts with a previous report suggesting that stimulation of NPVF neurons results in the inhibition of RN neurons (*Madelaine et al., 2017*). While the basis for the discrepancy between the two studies is unclear, Madelaine et al. used a different opsin, a different stimulation paradigm using visible light that can itself affect the activity of zebrafish RN neurons (*Cheng et al., 2016*), and a GCaMP line that, unlike the line used in our study, was not specifically expressed in serotonergic RN neurons, making it unclear precisely which neurons were analyzed. Despite this discrepancy, our finding that stimulation of NPVF neurons results in broad activation of anterior IRa neurons is consistent with our genetic, optogenetic, and chemogenetic behavioral data. Further studies are needed to explore the molecular and functional diversity of RN neurons in zebrafish and the role they play in sleep regulation.

The hypothalamus-hindbrain neuronal circuit that we have described can be integrated into a larger sleep-promoting network. We recently reported that epidermal growth factor receptor (EGFR) signaling is necessary and sufficient for normal sleep amounts in zebrafish, and that it promotes sleep, in part, via the NPVF system (*Lee et al., 2019*). We found that it does so by both promoting the expression of *npvf* and by stimulating *npvf*-expressing neurons. The EGFR ligands *egf* and *transforming growth factor alpha* are expressed in glial cells in the dorsal diencephalon, and *egfra*, the EGFR paralog that is primarily responsible for the role of EGFR signaling in sleep, is expressed in juxta-ventricular glial cells found along the hypothalamus, hindbrain, tectum, and cerebellum. Taken together with the current study, these results describe a genetic and neuronal circuit spanning EGFR signaling in glia, *npvf*-expressing neurons in the hypothalamus, and serotonergic RN neurons in the hindbrain. This pathway plays a key role in regulating sleep homeostasis, as inhibition of EGFR signaling or loss of 5-HT in RN neurons in zebrafish, and ablation of the dorsal and median RN in rodents, results in sleep homeostasis defects (*Lee et al., 2019*; *Oikonomou et al., 2019*).

If the EGFR-NPVF-RN sleep-promoting circuit plays a central and important role in regulating sleep, one might expect it to be evolutionarily conserved. Indeed, similar to zebrafish, EGFR signaling promotes sleep in *C. elegans* and *Drosophila* (*Donlea et al., 2009*; *Foltenyi et al., 2007*; *Konietzka et al., 2020*; *Van Buskirk and Sternberg, 2007*), and genetic experiments suggest that it does so in part via RFamide neuropeptides that may be invertebrate homologs of *npvf* (*He et al., 2013*; *Iannacone et al., 2017*; *Konietzka et al., 2020*; *Lenz et al., 2015*; *Nagy et al., 2014*; *Nath et al., 2016*; *Nelson et al., 2014*; *Shang et al., 2013*; *Turek et al., 2016*; *Van der Auwera et al., 2020*). Serotonin has also been shown to promote sleep in *Drosophila* (*Qian et al., 2017*;

*Yuan et al., 2006*), and by analogy to our results, we hypothesize that RFamide neuropeptides such as FMRFamide (*Lenz et al., 2015*) may act upstream of 5-HT to promote *Drosophila* sleep.

Evidence suggests that the EGFR-NPVF-RN sleep-promoting circuit likely extends to mammals. In humans, variation in genes that participate in EGFR signaling (*Lee et al., 2019*; *Wang et al., 2019*) and 5-HT signaling (*Dashti et al., 2019*; *Jones et al., 2019*; *Lane et al., 2019*; *Lane et al., 2017*) have been implicated by genome-wide association studies in human sleep traits and sleep disorders. In experimental organisms, intracerebroventricular injection of EGF in rabbits was sufficient to increase sleep (*Kushikata et al., 1998*), and mice containing linked mutations in *Egfr* and *Wnt3a* (*Wingless integration site 3a*) showed abnormal circadian timing of sleep (*Kramer et al., 2001*). Additionally, pharmacological inhibition or genetic loss of extracellular regulated kinase (ERK), which mediates EGFR signaling, resulted in reduced sleep in mice (*Mikhail et al., 2017*). The restricted pattern of NPVF expression in the medial hypothalamus is also conserved among human, rodent, and zebrafish brains (*Lee et al., 2017*; *Liu et al., 2001*; *Ubuka et al., 2009*; *Yelin-Bekerman et al., 2015*), as are the expression of EGFR and its ligands in zebrafish and rodents (*Lee et al., 2019*; *Ma et al., 1994*; *Ma et al., 1992*). However, NPVF has not been studied in the context of mammalian sleep, so further studies are required to determine whether the EGFR/NPVF/RN neural circuit described in zebrafish is fully conserved in mammals. Understanding the conserved role these systems play in controlling sleep and wakefulness in different animal models will provide insights into how sleep regulation has evolved (*Joiner, 2016*), and may reveal functions that this essential behavior engenders across animal phyla.

# Materials and methods

## Key resources table

| Reagent type (species) or resource | Designation | Source or reference | Identifiers | Additional information |
|---|---|---|---|---|
| Antibody | Rabbit polyclonal anti-5-HT | MilliporeSigma | Cat# S5545; RRID:AB_477522 | (1:1000) |
| Antibody | Chicken polyclonal anti-GFP | Aves Laboratory | Cat# GFP-1020, RRID:AB_10000240 | (1:1000) |
| Antibody | Rabbit polyclonal anti-DsRed | Takara Bio | Cat# 632496, RRID:AB_10013483 | (1:1000) |
| Antibody | Rabbit polyclonal anti-tagRFP | Evrogen | Cat# AB233, RRID:AB_2571743 | (1:200) |
| Antibody | Goat polyclonal anti-Chicken IgY (H+L) Secondary Antibody, Alexa Fluor 488 | Thermo Fisher Sci. | Cat# A-11039; RRID:AB_2534096 | (1:500) |
| Antibody | Goat polyclonal anti-Rabbit IgG (H+L) Cross-Adsorbed Secondary Antibody, Alexa Fluor 568 | ThermoFisher Sci. | Cat# A-11011; RRID:AB_143157 | (1:500) |
| Antibody | Goat polyclonal anti-Rat IgG (H+L) Cross-Adsorbed Secondary Antibody, Alexa Fluor 488 | ThermoFisher Sci. | Cat# A-11006, RRID:AB_2534074 | (1:500) |

*Continued on next page*

*Continued*

| Reagent type (species) or resource | Designation | Source or reference | Identifiers | Additional information |
|---|---|---|---|---|
| Chemical compound, drug | Metronidazole | MP Biomedicals | Cat# 0215571080 | |
| Strain, strain background (*Danio rerio*) | *npvf* ct845 mutant | *Lee et al., 2017* | RRID:ZDB-ALT-170927-1 | |
| Strain, strain background (*Danio rerio*) | *tph2* ct817 mutant | *Chen et al., 2013a* | RRID:ZDB-ALT-131122-14 | |
| Strain, strain background (*Danio rerio*) | *Tg(npvf:eGFP)* ct847Tg | *Lee et al., 2017* | RRID:ZDB-ALT-170927-3 | |
| Strain, strain background (*Danio rerio*) | *Tg(npvf:GCaMP6s-P2A-tdTomato)* ct872Tg | *Lee et al., 2019* | ZFIN: ZDB-ALT-190725–5 | |
| Strain, strain background (*Danio rerio*) | *Tg(npvf:Rea ChR-mCitrine)* ct849Tg | *Lee et al., 2017* | RRID:ZDB-ALT-170927-5 | |
| Strain, strain background (*Danio rerio*) | *Tg(npvf:kalta4)* ct848Tg | *Lee et al., 2017* | RRID:ZDB-ALT-170927-4 | |
| Strain, strain background (*Danio rerio*) | Zebrafish: *Tg(tph2:eNTR-mYFP)* ct866Tg | *Oikonomou et al., 2019* | RRID:ZDB-ALT-190508-3 | |
| Strain, strain background (*Danio rerio*) | *Tg(tph2:GCaMP6s-P2A-NLS:tdTomato)* ct874 | This study; *Figure 2* and *Figure 2—figure supplements 3* and *4*. | ZFIN: ZDB-ALT-200512–2 | *GCaMP6s-P2A-NLS:tdTomato* expressed under the *tph2* promoter; – Prober Lab |
| Strain, strain background (*Danio rerio*) | *Tg(UAS:nfsb-mCherry)* rw0144Tg | *Agetsuma et al., 2010* | RRID:ZDB-ALT-110215-7 | |
| Strain, strain background (*Danio rerio*) | *Tg(UAS:TRPV1-tagRFP-T)* ct851Tg | *Lee et al., 2017* | RRID:ZDB-ALT-170927-7 | |
| Sequence-based reagent | Primer: *tph2* mutant genotyping primer 1: AGAACTTACAAAACTCTATCCAACTC | *Oikonomou et al., 2019* | | |
| Sequence-based reagent | Primer: *tph2* mutant genotyping primer 2: AGAGAGGACAACATCTGGGG | *Oikonomou et al., 2019* | | |
| Sequence-based reagent | Primer: *tph2* mutant genotyping primer 3: TAATCATGCAGTCCGTTAATACTC | *Oikonomou et al., 2019* | | |
| Sequence-based reagent | Primer: *npvf* mutant genotyping primer 1: CAGTGGTGGTGCGAGTTCT | *Lee et al., 2017* | | |

*Continued*

| Reagent type (species) or resource | Designation | Source or reference | Identifiers | Additional information |
|---|---|---|---|---|
| Sequence-based reagent | Primer: *npvf* mutant genotyping primer 2: GCTGAG GGAGGTTGATGGTA | *Lee et al., 2017* | | |
| Sequence-based reagent | Primer: *Tg(npvf: ReaChR-mCitrine)* genotyping primer 1: CACGA GAGAATGCTGTTCCA | *Lee et al., 2017* | | |
| Sequence-based reagent | Primer: *Tg(npvf: ReaChR-mCitrine)* genotyping primer 2: CCATGG TGCGTTTGCTATAA | *Lee et al., 2017* | | |
| Sequence-based reagent | Primer: *Tg(UAS: TRPV1-tagRFP-T)* genotyping primer 1: CAGCCT CACTTTGAGCTCCT: | *Lee et al., 2017* | | |
| Sequence-based reagent | Primer: *Tg(UAS:TRPV1-tagRFP-T)* genotyping primer 2: TCCTCAT AAGGGCAGTCCAG | *Lee et al., 2017* | | |
| Software, algorithm | MATLAB R2017b | Mathworks | RRID:SCR_001622 | |
| Software, algorithm | Prism6 | GraphPad | RRID:SCR_002798 | |
| Software, algorithm | Image J/Fiji | *Schneider et al., 2012* | RRID:SCR_002285 | |
| Other | 96-well plate | GE Healthcare Life Sciences | Cat#: 7701–1651 | |
| Other | MicroAmp Optical Adhesive Film | Thermo Fisher Scientific | Cat#: 4311971 | |

## Experimental model and subject details

Animal husbandry and all experimental procedures involving zebrafish were performed in accordance with the California Institute of Technology Institutional Animal Care and Use Committee (IACUC) guidelines and by the Office of Laboratory Animal Resources at the California Institute of Technology (animal protocol 1580). All experiments used zebrafish on 5 and 6 dpf. Sex is not yet defined at this stage of development. Larvae were housed in petri dishes with 50 animals per dish. E3 medium (5 mM NaCl, 0.17 mM KCl, 0.33 mM $CaCl_2$, 0.33 mM $MgSO_4$) was used for housing and experiments. All lines were derived from the TLAB hybrid strain. Unless otherwise indicated, for experiments using mutant animals, heterozygous and homozygous mutant adult animals were mated, and their homozygous mutant and heterozygous mutant progeny were compared to each other, to minimize variation due to genetic background. For experiments using transgenic animals, heterozygous transgenic animals were outcrossed to non-transgenic animals of the parental TLAB strain, and transgenic heterozygous progeny were compared to their non-transgenic siblings. Behavioral experiments were performed blind to genotype, with animals genotyped by PCR after each experiment was complete.

## Transgenic and mutant animals

The *Tg(npvf:eGFP)* ct847Tg (*Lee et al., 2017*), *Tg(npvf:ReaChR-mCitrine)* ct849Tg (*Lee et al., 2017*), *Tg(npvf:GCaMP6s-P2A-tdTomato)* ct872Tg (*Lee et al., 2019*), *Tg(npvf:kalta4)* ct848Tg (*Lee et al., 2017*), *Tg(tph2:eNTR-mYFP)* ct866Tg (*Oikonomou et al., 2019*), *Tg(UAS:nfsb-mCherry)* rw0144Tg (*Agetsuma et al., 2010*), *Tg(UAS:TRPV1-tagRFP-T)* ct851Tg (*Lee et al., 2017*), *npvf* ct845 mutant

(*Lee et al., 2017*), and *tph2* ct817 mutant (*Chen et al., 2013a*) lines have been previously described. In the figures, *Tg(npvf:kalta4); Tg(UAS:TRPV1-tagRFP-T)* double transgenic and *Tg(UAS:TRPV1-tagRFP-T)* single transgenic animals are abbreviated as *npvf:TRPV1 +* and *npvf:TRPV1 -*, respectively.

To generate *Tg(tph2:GCaMP6s-P2A-NLS:tdTomato)* animals we cloned the *tph2* promoter (*Oikonomou et al., 2019*) upstream of cytoplasmic-localized GCaMP6s (*Chen et al., 2013b*) followed by an intein P2A sequence, which generates a self-cleaving peptide (*Kim et al., 2011*), and NLS-tdTomato. Stable transgenic lines were generated using the Tol2 method (*Urasaki et al., 2006*). This transgenic line is abbreviated to *Tg(tph2:GCaMP6s-P2A-tdTomato)* in the main text and figures.

## Immunohistochemistry

Samples were fixed in 4% paraformaldehyde/4% sucrose in PBS overnight at 4°C and then washed with 0.25% Triton X-100/PBS (PBTx). Immunolabeling was performed using dissected brains because this allows for superior antibody penetration. Dissected brains were incubated for 1 hr in 1 mg/mL collagenase (C9891, MilliporeSigma, St. Louis, Missouri, USA) and blocked overnight in 2% normal goat serum/2% DMSO in PBTx at 4°C. Incubation with rabbit anti-5-HT (1:1000; S5545, Millipore-Sigma, Burlington, MA, USA), chicken anti-GFP (1:1000, GFP-1020, Aves Laboratory, Davis, CA, USA), and rabbit anti-DsRed (1:1000, Takara Bio, Mountainview, CA, USA) primary antibodies was performed in blocking solution overnight at 4°C. Incubation with goat anti-rabbit IgG Alexa Fluor 568, goat anti-chicken IgY Alexa Fluor 488, and goat anti-rat IgG Alexa Fluor 488 (all 1:500, Thermo-Fisher Sci., Waltham, MA, USA) secondary antibodies was performed in blocking solution overnight at 4°C. Samples were mounted in Vectashield (H-1000; Vector Laboratories, Burlingame, CA, USA) and imaged using a Zeiss LSM 880 confocal microscope (Zeiss, Oberkochen, Germany).

## Two-photon optogenetic stimulation and GCaMP6s imaging

At 6 dpf, animals were paralyzed by immersion in 1 mg/ml α-bungarotoxin (2133, Tocris, Bristol, UK) dissolved in E3, embedded in 1.5% low melting agarose (EC-202, National Diagnostics, Atlanta, GA, United States) and imaged using a 20x water immersion objective on a Zeiss LSM 880 microscope equipped with a two-photon laser (Chameleon Coherent, Wilsonville, OR, USA) on a non-linear optics (NLO) anti-vibration table (Newport Instruments). Laser power coming out of the objective was quantified using a power meter (PM121D, ThorLabs, Newton, NJ, USA). For GCaMP6s imaging, a region of interest (ROI) that encompassed *npvf-* or *tph2*-expressing neuronal somas was defined based on nuclear localized tdTomato, which was equally co-expressed with GCaMP6s. GCaMP6s and tdTomato fluorescence intensity were quantified using Image J (*Schneider et al., 2012*). GCaMP6s fluorescence was normalized to tdTomato fluorescence to control for potential drift/movement artifacts and/or changes in transgene expression level over the long time interval of imaging. GCaMP6s and tdTomato fluorescence were excited using a 920 nm two-photon laser (Chameleon Coherent, Wilsonville, OR, USA) at 8 mW, imaged in a $512 \times 256$ pixel frame (1.27 s per frame, pixel size = 0.55 µm, pixel dwell time = 2.07 µs) for 150 frames to acclimate animals to the imaging paradigm. For optogenetic stimulation of NPVF neurons, a $150 \times 100$ pixel region that encompassed the NPVF neuronal somas was illuminated using the 920 nm two-photon laser at 38 mW. Ten pulses were applied over 3.72 s using the bleaching function at 2.7 Hz per pixel. The time between the final stimulation pulse and initiation of post-stimulation imaging was 0.6 s, and was due to the computer registering coordinate information with the scan device (~0.4 s) and for the non-descanned detector to turn off (~0.2 s). GCaMP6s and tdTomato fluorescence were then imaged again using 8 mW laser power for 150 frames before the next stimulation trial. For GCaMP6s imaging of NPVF and IRa neurons, GCaMP6s/tdTomato fluorescence intensity values were calculated for each neuron for each trial. Five *Tg(npvf:ReaChR); Tg(npvf:GCaMP6s-P2A-tdTomato)* and five *Tg(npvf:eGFP); Tg(npvf:GCaMP6s-P2A-tdTomato)* animals, with approximately 17 NPVF neurons per animal, were subjected to three optogenetic stimulation trials and analyzed for *Figure 2—figure supplements 1* and *2*. Four *Tg(npvf:ReaChR); Tg(tph2:GCaMP6s-P2A-tdTomato)* and four *Tg(npvf:eGFP); Tg(tph2:GCaMP6s-P2A-tdTomato)* animals, with approximately 30 IRa neurons analyzed per animal, were analyzed for *Figure 2* and *Figure 2—figure supplements 3* and *4*. Three optogenetic stimulation trials were performed on three fish, and two trials were performed on a fourth fish, for both genotypes. For purposes of visualization, all figures show twenty imaging frames pre- and post-

stimulation. Baseline GCaMP6s fluorescence ($F_0$) for each trial was defined as the average GCaMP6s/tdTomato value of each ROI from 20 imaging frames (~25 s) immediately before optogenetic stimulation. Post-stimulation fluorescence (F) values were quantified as the GCaMP6s/tdTomato value of each ROI for the average of 10 imaging frames immediately after optogenetic stimulation. $\Delta F/F_0$ was defined as $(F - F_0) / F_0$.

## Sleep/wake behavioral analysis

Sleep in zebrafish larvae is defined based on broadly-accepted behavioral criteria that include behavioral quiescence that is rapidly reversible, increased arousal threshold, and a homeostatic response to sleep deprivation (*Campbell and Tobler, 1984*). Several labs have shown that zebrafish exhibit behavioral states that meet these criteria (*Prober et al., 2006*; *Yokogawa et al., 2007*; *Zhdanova et al., 2001*). In larval zebrafish, one or more minutes of inactivity is associated with an increased arousal threshold, and can thus be defined as a sleep state (*Elbaz et al., 2012*; *Prober et al., 2006*). Sleep/wake analysis was performed as previously described (*Prober et al., 2006*). Larvae were raised on a 14:10 hr light:dark (LD) cycle at 28.5°C with lights on at 9 a.m. and off at 11 p.m. Dim white light was used to raise larvae for optogenetic experiments to prevent stimulation of ReaChR by ambient light during development. Individual larvae were placed into each well of a 96-well plate (7701–1651, Whatman, Pittsburgh, PA, United States) containing 650 µl of E3 embryo medium. Locomotor activity was monitored using a videotracking system (Viewpoint Life Sciences, Lyon, France) with a Dinion one-third inch Monochrome camera (Dragonfly 2, Point Grey, Richmond, Canada) fitted with a variable-focus megapixel lens (M5018-MP, Computar, Cary, NC, United States) and infrared filter. The movement of each larva was recorded using the quantization mode. The 96-well plate and camera were housed inside a custom-modified Zebrabox (Viewpoint Life Sciences) that was continuously illuminated with infrared light. The 96-well plate was housed in a chamber filled with recirculating water to maintain a constant temperature of 28.5°C. Data were analyzed using custom Perl and Matlab (Mathworks, Natick, MA, United States) scripts (*Lee et al., 2017*), which conform to the open source definition.

## Optogenetic stimulation

Optogenetic behavioral experiments were performed as described (*Singh et al., 2015*). These experiments use a videotracking system with a custom array containing three sets of blue LEDs (470 nm, MR-B0040-10S, Luxeon V-star, Brantford, Canada) mounted 15 cm above and 7 cm away from the center of the 96-well plate to ensure uniform illumination. The LEDs were controlled using a custom-built driver and software written in BASIC stamp editor. A power meter (1098293, Laser-check, Santa Clara, CA, USA) was used before each experiment to verify uniform light intensity (~800 µW $cm^{-2}$ at the surface of the 96-well plate). In the afternoon of the fifth day of development, single larvae were placed into each well of a 96-well plate and placed in a videotracker in the dark. Larvae were exposed to blue light for 30 min for each of three trials at 12:30 am, 3:00 am, and 5:30 am. Behavior was monitored for 30 min before and after light onset. Light onset induces a startle response, which causes a short burst of locomotor activity. For this reason, we excluded 5 min of behavioral recording centered at the peak of blue light onset from analysis. Data was normalized by dividing the locomotor activity or sleep of each animal during light exposure by the average baseline locomotor activity or sleep of all animals of the same genotype. For baseline, we used a time period equal in length to blue light exposure, but prior to light onset.

## Chemogenetic ablation

Animals were treated with 5 mM metronidazole (MTZ) (0215571080, MP Biomedicals, Santa Ana, CA, USA) diluted in E3 medium containing 0.1% DMSO, starting in the afternoon at 2 dpf, and refreshed every 24 hr. Animals were kept in dim light during the day to prevent MTZ photodegradation. On the evening at 4-dpf, the animals were rinsed three times in E3 medium, allowed to recover for ~60 min, and then transferred to 96-well plates. Reported data is from the 5th day and night of development.

## Chemogenetic stimulation

Neuronal stimulation using TRPV1 was performed as described (*Lee et al., 2017*) with some modifications. Capsaicin (M2028, Sigma, St. Louis, Missouri, USA) was dissolved in DMSO to prepare a 100 mM stock solution that was stored in aliquots at −20°C. Capsaicin working solutions were prepared just before each experiment by diluting the stock solution in E3 medium. Larvae were placed into 96-well plates immersed in either 2 µM capsaicin or DMSO vehicle starting on the afternoon of 4 dpf as previously described (*Lee et al., 2017*; *Ly et al., 2020*). All treatments contained a final concentration of 0.002% DMSO. Behavioral analysis was performed from 5 dpf until 6 dpf.

## Quantification and statistical analysis

For all behavioral experiments, the unit of analysis for statistics is a single animal. For GCaMP6s imaging experiments, the unit of analysis for statistics is the GCaMP6s/tdTomato fluorescence value for a single neuron for a single optogenetic stimulation trial. The number of neurons or animals analyzed are either shown in the figure or stated in the figure legend. Behavioral traces (line graphs) represent mean and were generated from normalized optogenetic data (*Figures 5A,B* and *6G, H*, and *Figure 5—figure supplement 1A,B*) or raw data that was smoothed over 1 hr bins in 10 min intervals (*Figures 3A–D*, *4I–L* and *6A–D*). The significance threshold was set to $p < 0.05$ unless otherwise specified, and p-values were adjusted for multiple comparisons where appropriate. For one-factor design datasets that were not normally distributed, as assessed by D'Agostino and Pearson omnibus normality test, a non-parametric statistical test (Mann-Whitney test for two unpaired groups) was used as previously described (*Chiu et al., 2016*; *Lee et al., 2017*). For one-factor design test statistics that follow a normal distribution among two comparison groups, we applied a two-tailed Student's t-test, or a one-sample t test where appropriate. For comparison of differences between groups with two-factor designs, we used Two-Way ANOVA with Holm-Sidak test for multiple comparisons (*Figures 3E*, *4M,N* and *6E,F* and *Figure 2—figure supplement 3M*). For box plots, the box extends from the 25th to the 75th percentile with the median marked by a horizontal line through the box. The lower and upper whiskers extend to the 10th and 90th percentile, respectively. Data points outside the lower and upper whiskers were not shown in the graphs to facilitate data presentation but were included in statistical analyses. Statistical analyses were performed using Prism 6 (GraphPad Software, San Diego, CA, USA).

## Source code availability

The source code used for data analysis is available at https://elifesciences.org/articles/25727 (*Lee et al., 2017*).

## Acknowledgements

We thank members of the Prober lab for helpful discussions; Sarah Hou for experimental assistance; Uyen Pham, Chris Cook, Caressa Wong, Axel Dominguez and Alex Mack for zebrafish husbandry assistance; and Andres Collazo, Giada Spigolon, and the Beckman Institute Biological Imaging Facility for 2-photon imaging assistance. This work was supported by grants from the NIH (DAL: K99NS097683, F32NS084769; GO: F32NS082010; DAP: NS070911, NS101158), a NARSAD Young Investigator Grant (DAL: 25392) and a Caltech BBE Postdoctoral Fellowship to DAL. The authors declare no competing interests.

## Additional information

### Funding

| Funder | Grant reference number | Author |
| --- | --- | --- |
| National Institute of Neurological Disorders and Stroke | K99NS097683 | Daniel A Lee |
| National Institute of Neurological Disorders and Stroke | F32NS084769 | Daniel A Lee |
| Brain and Behavior Research | NARSAD Grant: 25392 | Daniel A Lee |

| | | |
|---|---|---|
| Foundation | | |
| California Institute of Technology | BBE Divisional Postdoctoral Fellowship | Daniel A Lee |
| National Institute of Neurological Disorders and Stroke | F32NS082010 | Grigorios Oikonomou |
| National Institute of Neurological Disorders and Stroke | NS070911 | David A Prober |
| National Institute of Neurological Disorders and Stroke | NS101158 | David A Prober |

The funders had no role in study design, data collection and interpretation, or the decision to submit the work for publication.

### Author contributions

Daniel A Lee, Conceptualization, Resources, Data curation, Formal analysis, Supervision, Funding acquisition, Validation, Investigation, Visualization, Methodology, Writing - original draft, Project administration, Writing - review and editing; Grigorios Oikonomou, Resources, Software, Methodology, Writing - review and editing; Tasha Cammidge, Young Hong, Data curation, Formal analysis, Investigation; Andrey Andreev, Resources, Data curation, Formal analysis, Validation, Investigation, Visualization, Methodology; Hannah Hurley, Formal analysis, Investigation; David A Prober, Conceptualization, Resources, Formal analysis, Supervision, Funding acquisition, Project administration, Writing - review and editing

### Author ORCIDs

Daniel A Lee https://orcid.org/0000-0001-7411-2740
Grigorios Oikonomou http://orcid.org/0000-0001-6797-7375
Andrey Andreev https://orcid.org/0000-0002-7833-1390
Young Hong https://orcid.org/0000-0001-9548-5511
David A Prober https://orcid.org/0000-0002-7371-4675

### Ethics

Animal experimentation: This study was performed in strict accordance with the recommendations in the Guide for the Care and Use of Laboratory Animals of the National Institutes of Health. All experiments were performed using standard protocols (Westerfield, 1993) in accordance with the California Institute of Technology Institutional Animal Care and Use Committee guidelines and by the Office of Laboratory Animal Resources at the California Institute of Technology (animal protocol 1580).

### Decision letter and Author response

Decision letter https://doi.org/10.7554/eLife.54491.sa1
Author response https://doi.org/10.7554/eLife.54491.sa2

## Additional files

### Supplementary files

• Transparent reporting form

### Data availability

All data generated or analyzed during this study are included in the manuscript and supporting files. Details described in this paper regarding transgenic and mutant animals have been deposited at ZFIN.

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
