## [Decision Letter]

**Decision letter after peer review:**

Thank you for submitting your article "Neuropeptide VF neurons promote sleep via the serotonergic raphe" for consideration by *eLife*. Your article has been reviewed by Catherine Dulac as the Senior Editor, a Reviewing Editor, and two reviewers. The following individual involved in review of your submission has agreed to reveal their identity: Henrik Bringmann (Reviewer #2).

The reviewers have discussed the reviews with one another and the Reviewing Editor has drafted this decision to help you prepare a revised submission.

Summary:

In this manuscript submitted as a Research Advance, the authors build on their prior *eLife* research article centered on the role of NPVF in sleep regulation in zebrafish. Here, they show that some hypothalamic NPVF neurons project to serotonergic neurons in the brain stem raphe. They combine these anatomic studies with functional evidence that NPVF neurons, via action of NPVF, are (usually) excitatory to serotonergic raphe neurons and with behavioral evidence suggesting that the quiescence defects resulting from genetic removal of the npvf gene are fully explained by effects on serotonergic transmission. While highly focused, this article represents a significant advance in our understanding of the circuitry regulating sleep/wake behavior in vertebrates.

Essential revisions:

The reviewers felt that there were a couple areas that needed attention before the paper was ready to be published.

1) The authors need to more fully discuss their work in the context of the relevant literature. Specifically, their experiments predict that NPVF receptor(s) is(are) expressed in the raphe. Three articles (one in fish PMID 28139691, and two in rodents, PMID 11024015 and PMID 12932818) show that GPR147 in fact is expressed in the vertebrate raphe. They should discuss the receptor expression and cite these papers. Also, Figure 2E: there is a large difference between these results and those shown in Figure 3B-C of Madelaine et al. Since this manuscript concludes an opposite sign (excitatory) from Madelaine et al., (inhibitory) for the npvf-raphe synapse, it is important to discuss possible explanation for these contrasting results. Along these lines, were the 38 iRa neurons that reduced GCaMP signal with npvf neuron stimulation anatomically segregated from the 116 neurons that showed an excitatory response? Could site of imaging explain the discrepant results with those of Madelaine et al.?

2) Reviewers were also concerned about some of the optogenetic data. This concern could be addressed experimentally in the revision or a more complete discussion of the caveats of interpretation should be provided. The major concern is Figure 5: a combination of optogenetic tool and blue light is used to "induce" sleep. However, the blue light stimulus induces activity and suppresses sleep. NPVF ReaChR attenuates this effect. The interpretation is that NPVF ReaChR induces sleep, but also other interpretations could be possible, such as an inhibition of the blue light response independently of sleep. This experiment is thus difficult to interpret. Would it not be possible to use a different, higher, wavelength (i.e. ReaChR could also be activated by green or orange light)? In one of the experiments, the authors used two-photon excitation to trigger optogenetic activation. Would this also be possible for this behavioral experiment and would this avoid the arousing light response? This experiment would be convincing if the optogenetic stimulation was set up to indeed increase sleep over baseline. If this is not possible technically then this issue should at least be discussed.

---

## [Author Response]

Essential revisions:The reviewers felt that there were a couple areas that needed attention before the paper was ready to be published.1) The authors need to more fully discuss their work in the context of the relevant literature. Specifically, their experiments predict that NPVF receptor(s) is(are) expressed in the raphe. Three articles (one in fish PMID 28139691, and two in rodents, PMID 11024015 and PMID 12932818) show that GPR147 in fact is expressed in the vertebrate raphe. They should discuss the receptor expression and cite these papers.

We thank the reviewers for pointing out our omission. While we did reference two of these papers in the original submission, we have added further discussion of NPVF receptor expression from all three papers throughout the revised manuscript (Bonini et al., 2000; Madelaine et al., 2017; Roumy et al., 2003).

2) Also, Figure 2E: there is a large difference between these results and those shown in Figure 3B-C of Madelaine et al. Since this manuscript concludes an opposite sign (excitatory) from Madelaine et al., (inhibitory) for the npvf◊raphe synapse, it is important to discuss possible explanation for these contrasting results.

We have added additional discussion in the manuscript and responses below to address these contrasting results.

Along these lines, were the 38 iRa neurons that reduced GCaMP signal with npvf neuron stimulation anatomically segregated from the 116 neurons that showed an excitatory response?

In order to address this comment, we re-analyzed the *Tg(npvf:ReaChR-mCitrine); Tg(tph2:GCaMP6s-P2A-tdTomato)* dataset to more rigorously evaluate positive and negative IRa neuron GCaMP6s responses to stimulation of NPVF neurons, and to determine whether there was an anatomical segregation to the magnitude and direction of IRa responses (Figure 2 and Figure 2—figure supplements 3 and 4). Since the anterior half of the IRa is densely innervated by NPVF neurons but the posterior half is not (Figure 1 and Figure 1—figure supplement 1), we hypothesized that stimulation of NPVF neurons would primarily activate neurons in the anterior IRa. Consistent with this hypothesis, we found that most IRa neurons that showed a large increase in normalized GCaMP6s fluorescence in response to stimulation of NPVF neurons were in the anterior IRa (Figure 2—figure supplement 3C). IRa neurons that responded with a small decrease in normalized GCaMP6s fluorescence were observed in both the anterior and posterior IRa (Figure 2—figure supplement 3D), although these were more frequently observed in the posterior IRa. Importantly, these small decreases in normalized GCaMP6s fluorescence were similar to those observed in *Tg(npvf:eGFP)* controls (Figure 2—figure supplement 3B). Normalized GCaMP6s fluorescence was not significantly changed in either the anterior or posterior IRa in *Tg(npvf:eGFP)* controls (Figure 2—figure supplements 3 and 4A), with most neurons showing either a small increase or a small decrease in normalized GCaMP6s fluorescence that together did not result in a statistically significant change (Figure 2—figure supplement 3A,B,M). We conclude that the predominant effect of stimulating NPVF neurons on the IRa is to activate neurons in the anterior IRa, and there is little or no evidence that stimulation of NPVF neurons results in inhibition of any IRa neurons.

We note that in our re-analysis we identified small motion artifacts (2-5 μm) between some optogenetic stimulation trials of both *Tg(npvf:eGFP)* and *Tg(npvf:reaChR-mCitrine)* animals. This affected our ability to track some of the neurons in successive imaging trials. As a result, in our new analysis we treat each neuron in each optogenetic trial as a separate data point, with GCaMP6s fluorescence normalized to tdTomato fluorescence for each neuron.

We now also show GCaMP6s data for each animal in Figure 2 and Figure 2—figure supplements 3 and 4.

Could site of imaging explain the discrepant results with those of Madelaine et al.?

We are unsure of the basis for the discrepancy, but there are several possibilities:

1) Site of imaging. The larval zebrafish serotonergic raphe have been described as subdivided into what are termed the superior and inferior raphe (Lillesaar et al., 2009). The inferior raphe is located caudal and ventral to the superior raphe (Figure 1E-K). Using the *tph2* promoter to regulate expression of GCaMP6s, we specifically imaged serotonergic neurons of the inferior raphe. The Madelaine et al. study imaged neurons in what was termed the “ventral raphe nucleus (vRN)”, which they described as the “ventral-posterior part of the raphe nucleus". It is unclear how the neurons imaged by Madelaine et al. correspond to those imaged in our study.

2) PTU treatment. Madelaine et al. used PTU treatment to suppress the production of body pigmentation, but PTU can have deleterious effects on development and physiology (Li et al., 2012) that may affect optogenetic stimulation and/or GCaMP imaging. We instead used *nacre* mutants (White et al., 2008) that lack most body pigment at larval stages, but retain a pigmented retinal epithelium, and thus retain a functional visual system, and are healthier than PTU-treated animals.

3) Different GCaMP transgenic lines. We used *Tg(tph2:GCaMP6s-P2A-tdTomato)* zebrafish in which cytoplasmic-localized GCaMP6s is specifically expressed in serotonergic neurons in the superior and inferior raphe. Madelaine et al. used *Tg(elavl3:h2b-GCaMP6s)* animals in which most neurons in the brain express nuclear localized GCaMP6s, and imaged neurons in what was termed the “ventral raphe (vRN)”. Thus, the neurons analyzed by Madelaine et al. were based on anatomical location rather than on a molecular marker, and might consist of previously reported nonserotonergic (Lillesaar et al., 2009) and/or non-raphe neurons that could have different properties from the molecularly defined serotonergic raphe neurons analyzed in our study.

4) Different imaging approaches. Since a previous study found that visible light can inhibit serotonergic raphe neurons (Cheng et al., 2016), we used invisible 920 nm two-photon light both at high power to stimulate ReaChR in NPVF neurons, and at low power to excite GCaMP6s fluorescence in IRa neurons. In contrast, Madelaine et al. used “full-field illumination through the imaging objective with a 588 nm laser”. This light is visible to the animals, and thus may confound studies of GCaMP fluorescence in the raphe.

5) Motion artifact. The Madelaine et al. study used non-paralyzed animals that were head restrained in agarose, but with the tail free to move. A concern with this approach is the possibility of motion artifacts that change the plane of focus, resulting in an apparent decrease in GCaMP fluorescence, which could explain their observations. This is particularly worrisome because, unlike our study, they did not co-express a second fluorescent protein along with GCaMP in order to control for possible motion artifacts. In order to eliminate this potential confound, we used α-bungarotoxin to paralyze the animals, we fully embedded the animals in agarose to increase the stability of the preparation, and we co-expressed nuclear-localized tdTomato with GCaMP6s and quantified the ratio of GCaMP6s to tdTomato fluorescence.

6) Different opsin. Madelaine et al. used C1V1 while we used ReaChR. These are both red-shifted variants of ChR2, but they have different properties. ReaChR photostimulation results in approximately four times more photocurrent than C1V1 photostimulation using either 470 nm or 590 nm light (Lin et al., 2013).

7) Laser power. In our study, the soma of NPVF neurons were illuminated using 10 pulses of a 920 nm two-photon laser at 38 mW (detected by a power meter placed on the sample) applied over 3.72 seconds. The Madelaine et al. study used a “588 nm laser at 80% output for ~3 s”, but did not describe the laser power used. Stimulation of opsins with intense light can lead to depolarization block as a result of insufficient repolarization (Lin et al., 2013). If Madelaine et al. used a laser intensity that was sufficient to induce depolarization block, this could account for the discrepant results between the two studies.

3) Reviewers were also concerned about some of the optogenetic data. This concern could be addressed experimentally in the revision or a more complete discussion of the caveats of interpretation should be provided. The major concern is Figure 5: a combination of optogenetic tool and blue light is used to "induce" sleep. However, the blue light stimulus induces activity and suppresses sleep. NPVF ReaChR attenuates this effect. The interpretation is that NPVF ReaChR induces sleep, but also other interpretations could be possible, such as an inhibition of the blue light response independently of sleep. This experiment is thus difficult to interpret. Would it not be possible to use a different, higher, wavelength (i.e. ReaChR could also be activated by green or orange light)? In one of the experiments, the authors used two-photon excitation to trigger optogenetic activation. Would this also be possible for this behavioral experiment and would this avoid the arousing light response? This experiment would be convincing if the optogenetic stimulation was set up to indeed increase sleep over baseline. If this is not possible technically then this issue should at least be discussed.

In contrast to the GCaMP6s imaging experiment in which single zebrafish are embedded in agarose and mounted on a two-photon microscope for optogenetic stimulation and GCaMP6s imaging, the optogenetic behavioral experiment monitors the behavior of 96 freely-behaving animals in a 96-well plate, and is not amenable to two-photon activation across such a large area. In order to address this comment we repeated the experiment using a red light stimulus as suggested by the reviewers, as ReaChR can induce photocurrents in response to both blue and red light (Lin et al., 2013). We observed that red light stimulated locomotor activity and suppressed sleep in WT animals to an extent similar to blue light at comparable intensity, and still had an arousing effect even when we decreased red light intensity by 90%. The red light stimulus caused decreased locomotor activity and increased sleep in *Tg(npvf:ReaChR)* animals compared to WT siblings, similar to the blue light stimulus. However, we observed a stronger phenotype using blue light, consistent with reports that ReaChR can induce action potentials ~ 50x more robustly in response to blue light compared to red light (Lin et al., 2013). Since we observed a stronger phenotype using blue light, we have used this data in the paper.

We note that using a fiber optic to deliver light into the mammalian brain can affect neuronal activity and behavior in WT animals (Hirase et al., 2002). Similarly, many *Drosophila* sleep studies use thermogenetic stimulation of specific neurons using heterologously expressed TRPA1, yet the increased temperature used to stimulate TRPA1-expressing neurons has profound effects on the behavior of non-TRPA1-expressing control animals. Thus, we suggest that interpretation of our optogenetic data is associated with the same caveat as rodent optogenetic and *Drosophila* thermogenetic studies, but is a reasonable approach when WT controls are subjected to the same stimulus as experimental animals.

However, as an alternative approach that avoids the potentially confounding effect of a light stimulus on behavior, in the original submission we also used a chemogenetic approach to stimulate NPVF neurons to test the hypothesis that NPVF neuron-induced sleep requires 5-HT in RN neurons. We previously generated *Tg(npvf:KalTA4)*; *Tg(UAS:TRPV1-TagRFP-T)* zebrafish (Lee et al., 2017), in which an optimized version of the transcriptional activator Gal4 drives expression of the rat TRPV1 ion channel (Chen et al., 2016) specifically in NPVF neurons. Using these transgenic animals, we previously showed that addition of the TRPV1 small molecule agonist capsaicin to the water results in specific stimulation of TRPV1-expressing NPVF neurons and increased sleep at night (Lee et al., 2017). In the original submission we reproduced our published result that chemogenetic stimulation of *npvf-*expressing neurons induces sleep at night, and showed that this effect is blocked in *tph2* mutant animals (which lack 5-HT in RN neurons) (Figure 6A-F in the revised manuscript), consistent with our interpretation of the optogenetic data.

As additional confirmation of our optogenetic data, we have added a new experiment in which we use TRPV1-mediated chemogenetic stimulation to test the hypothesis that NPVF neuron-induced sleep requires serotonergic RN neurons (Figure 4). To do so, we chemogenetically stimulated *npvf*-expressing neurons using TRPV1 in animals in which the RN were chemogenetically ablated using enhanced nitroreductase (eNTR) (Mathias et al., 2014; Tabor et al., 2014). We previously showed that ablation of RN neurons in *Tg*(*tph2*:*eNTR-mYFP*) animals results in decreased sleep (Oikonomou et al., 2019). This phenotype is similar to those of both *tph2* -/- zebrafish and to mice in which the dorsal and median serotonergic RN are ablated (Oikonomou et al., 2019). Similar to our previous report (Oikonomou et al., 2019), treatment of *Tg*(*tph2*:*eNTR-mYFP*) animals with 5 mM MTZ during 2-4 dpf resulted in near complete loss of YFP fluorescence and 5-HT immunoreactivity in the RN (Figure 4E,F). In contrast, treatment of these animals with DMSO vehicle control (Figure 4C,D), or treatment of *Tg*(*tph2*:*eNTRmYFP*) negative siblings with MTZ (Figure 4G,H), did not cause loss of the RN. Consistent with our previous observation (Lee et al., 2017), *Tg(npvf:KalTA4); Tg(UAS:TRPV1-tagRFP-T*) animals treated with both MTZ and capsaicin showed an increase in nighttime sleep compared to their identically treated *Tg(npvf:KalTA4)* siblings in *Tg(tph2:eNTR-mYFP)* negative animals (Figure 4I-N). In contrast, there was no significant difference in nighttime sleep between *Tg(npvf:KalTA4); Tg(UAS:TRPV1-tagRFP-T*); *Tg(tph2:eNTR-mYFP)* animals treated with both MTZ and capsaicin, and their identically treated *Tg(npvf:KalTA4)*; *Tg(tph2:eNTR-mYFP)* siblings (Figure 4I-N). This result indicates that sleep induced by chemogenetic stimulation of NPVF neurons requires serotonergic RN neurons, consistent with our interpretation of the optogenetic experiment in the original submission. We suggest that the two stimulation methods (optogenetics and chemogenetics) provide compelling evidence that stimulation of NPVF neurons promotes sleep in a manner that requires 5-HT in RN neurons.